# Advancing Pathogen Identification: The Role of Digital PCR in Enhancing Diagnostic Power in Different Settings

**DOI:** 10.3390/diagnostics14151598

**Published:** 2024-07-25

**Authors:** Alessia Mirabile, Giuseppe Sangiorgio, Paolo Giuseppe Bonacci, Dalida Bivona, Emanuele Nicitra, Carmelo Bonomo, Dafne Bongiorno, Stefania Stefani, Nicolò Musso

**Affiliations:** 1Department of Biomedical and Biotechnological Sciences (BIOMETEC), University of Catania, 95125 Catania, Italy; alessiamirabile93@gmail.com (A.M.); giuseppe.sangiorgio@phd.unict.it (G.S.); paolo.bonacci@phd.unict.it (P.G.B.); dalida.bivona@phd.unict.it (D.B.); emanuele.nicitra@phd.unict.it (E.N.); carmelo.bonomo@phd.unict.it (C.B.); stefania.stefani@unict.it (S.S.); nmusso@unict.it (N.M.); 2U.O.C. Laboratory Analysis Unit, University Hospital Policlinico-San Marco, Via Santa Sofia 78, 95123 Catania, Italy

**Keywords:** digital PCR, infectious disease diagnostics, environmental surveillance

## Abstract

Digital polymerase chain reaction (dPCR) has emerged as a groundbreaking technology in molecular biology and diagnostics, offering exceptional precision and sensitivity in nucleic acid detection and quantification. This review highlights the core principles and transformative potential of dPCR, particularly in infectious disease diagnostics and environmental surveillance. Emphasizing its evolution from traditional PCR, dPCR provides accurate absolute quantification of target nucleic acids through advanced partitioning techniques. The review addresses the significant impact of dPCR in sepsis diagnosis and management, showcasing its superior sensitivity and specificity in early pathogen detection and identification of drug-resistant genes. Despite its advantages, challenges such as optimization of experimental conditions, standardization of data analysis workflows, and high costs are discussed. Furthermore, we compare various commercially available dPCR platforms, detailing their features and applications in clinical and research settings. Additionally, the review explores dPCR’s role in water microbiology, particularly in wastewater surveillance and monitoring of waterborne pathogens, underscoring its importance in public health protection. In conclusion, future prospects of dPCR, including methodological optimization, integration with innovative technologies, and expansion into new sectors like metagenomics, are explored.

## 1. Introduction

In the realm of molecular biology and diagnostics, the emergence of digital polymerase chain reaction (dPCR) has sparked a revolution in the way we perceive and manipulate nucleic acids. This novel technique offers unparalleled precision and sensitivity, enabling researchers and clinicians to detect and quantify nucleic acids with unprecedented accuracy. As we delve into the intricate world of dPCR, it becomes evident that its applications extend far beyond traditional PCR methodologies. dPCR represents a technological leap from conventional polymerase chain reaction methods, offering an accurate absolute quantification of target nucleic acid molecules present in a sample [1]. This advanced technique partitions the sample into numerous compartments within which nucleic acids undergo stochastic partitioning followed by clonal amplification [2]. Concentrations are determined by analyzing the proportion of nonfluorescent particles according to the Poisson distribution. Recent advancements have led to the development of several commercial PCR droplet platforms, aimed at expediting the clinical adoption of digital PCR.

This review aims to provide an overview of dPCR, focusing on its applications in infectious disease diagnostics and environmental surveillance. We will delve into the fundamental principles underlying dPCR, highlighting its unique ability to partition nucleic acids into individual reactions, thereby facilitating absolute quantification. Beyond its technical intricacies, the true essence of dPCR lies in its transformative potential across various fields. From precision medicine to environmental monitoring, dPCR offers a versatile toolkit for addressing complex biological questions and tackling real-world challenges.

As we navigate through the potential applications, it is imperative to address the challenges, the limitations, and the future perspective in dPCR technology. From optimizing experimental conditions to standardizing data analysis workflows, there are several obstacles that must be overcome to fully realize the potential of dPCR in clinical and research settings. By critically evaluating these challenges, we can chart a course toward harnessing the full power of dPCR for advancing science and improving healthcare outcomes.

## 2. Digital PCR: Brief History and Concept

In recent decades, molecular technologies have evolved significantly. Classic PCR, considered a first-generation amplification method, has paved the way for newer PCR-based technologies. Classic PCR allows in vitro nucleic acid sequence detection using specific oligodeoxynucleotides as primers. Upon annealing to their targets, a thermostable DNA polymerase amplifies millions of copies, termed amplicons [3]. The PCR reaction involves alternating temperature cycles, typically repeated 20 to 40 times, facilitating denaturation of DNA duplex, primer hybridization to the target sequence, and elongation by DNA polymerase enzyme. In every cycle, the target number doubles; after n cycles, the exponential amplification reaction yields 2^n^ copies [2]. However, as an endpoint technique, classic PCR necessitates post-PCR methods like agarose gel electrophoresis for qualitative analysis [4]. The second-generation method includes both qualitative and quantitative analysis, featuring real-time detection of the amplification reaction, named quantitative real-time PCR (qRT-PCR). This fluorescence-based technology measures emitted fluorescence to indicate the target presence and its concentration. To increase accuracy, relative quantification is measured in comparison with a standard curve, derived from a sample of known concentration, ensuring precise results [5]. Due to its reproducibility and simplicity, qRT-PCR remains the gold standard for nucleic acid detection in clinical settings today. While used and advantageous, some of the limitations of qRT-PCR are addressed by third-generation PCR-based technologies. These methods enable target quantity detection without internal control and reduce PCR inhibitors’ exposure. In addition, deep target detection is guaranteed by partitioning the sample into a thousand individual PCRs. These features characterize dPCR, which, being a highly sensitive instrument, is applied to various molecular analyses, even those associated with rare nucleic acid variations [6]. Initially, this technique was referred to as “single-molecule PCR” or “dilution PCR”, the name dPCR was coined by Vogelstein and Kinzler in 1988 [7,8]. Over time, an implementation of the technology was pursued; indeed, a large reagent volume was required for a small number of partitions, resulting in high-cost reactions [9]. To date, a large number of partitions in a reduced space allows for working with a low reagent volume, decreasing the reaction costs. The main characteristic of the dPCR is the ability to detect DNA/RNA for a large-scale amplification starting from a single-molecule template. Furthermore, each individual target molecule undergoes an individual PCR reaction within chambers, wells, or droplets depending on the type of dPCR [10]. Additionally, the concentration of the volume reaction into microreactors enables target compartmentation, thereby obtaining partitions with either few or no target sequences. From each positive partition, a fluorescent signal indicates the detection of the target, and the digital reading assigns the number 1 to them. Conversely, the digital reading associates the number 0 with the partitions that show no fluorescence emissions [11,12]. The ratio between PCR-positive and PCR-negative partitions determines the total amount of the target based on the Poisson’s distribution. The most significant benefit of this compartmentalization process is the absence of target competition, allowing for the easy detection of rare molecules amidst a background of wild-type sequences. While dPCR offers several advantages over traditional qPCR methods, it is improbable that it will completely supplant qPCR assays in clinical laboratory settings. Indeed, high-cost consumables, a laborious data analysis process, and the need for specialized personnel are required. The current lower throughput and longer turnaround times of dPCR systems compared to qPCR may raise doubts about their routine implementation for now. In addition, any loss of volume during the partitioning process can bias the final target calculations. Moreover, many issues arise for highly concentrated targets; proper dilutions of the samples are fundamental to avoid signal saturation and obtain a quantifiable result [13]. Despite improvements in multiplexing capability using dPCR, interference between the different optical channels and trouble separating and clustering partitions make the analysis difficult [14]. Additionally, certain factors leading to bias and variation, such as DNA denaturation during partitioning, can result in the splitting of single strands into two distinct units, leading to an overestimation. Conversely, underestimation may occur due to factors like “molecular dropout”, template linkage, sample inconsistency, and variance in partition volume [15].

## 3. Commercially Available Digital Platform

In the context of nucleic acid quantification, digital PCR (dPCR) and droplet digital PCR (ddPCR) stand out as revolutionary techniques, offering unparalleled precision and sensitivity. While both methodologies fall under the umbrella of digital PCR, they diverge in their approach to partitioning the sample and analyzing the amplification products (Figure 1). Understanding the nuances between dPCR and ddPCR is essential for researchers aiming to harness the full potential of these cutting-edge technologies in various fields, from genetic analysis to diagnostics. When dPCR and ddPCR are compared, the partitioning strategy is the primary distinction (Table 1).

### 3.1. Droplet Digital PCR (ddPCR)

A droplet digital PCR system produces tens of thousands of submicroliter droplets from an immiscible fluid in oil [17,18]. Droplets function as miniature reaction chambers, encasing random nucleic acids. The digital droplet PCR reaction is made in a tube in a typical ddPCR workflow. A droplet generator is then used to separate the ddPCR mixture into discrete droplets. After gathering the emulsion in a vial, endpoint PCR is carried out. Droplets are fluorescently read one by one as they pass in front of a laser excitation source in a flow cytometer, which processes the sample. Poisson distribution can be used to determine the number of copies of DNA template in the initial reaction, just like with any other dPCR protocol. Currently, the platforms performing droplet digital PCR include the following:

#### 3.1.1. Bio-Rad QX200™ Droplet Digital PCR System (Bio-Rad Laboratories S.r.l, Milan, Italy)

This is one of the earliest and most widely used ddPCR platforms. It allows precise and absolute quantification of target nucleic acids, offering applications in various fields including oncology, infectious diseases, and genetic testing [19]. Using this technology, Liu et al. [20] developed a highly sensitive ddPCR assay to detect MYU and AP3B1 in urine samples for potential use in diagnosing and evaluating prostate cancer (PCa). Comparing ddPCR and qPCR, they compared repeated measurements on copy numbers ranging from 6.7 up to 67,000, demonstrating ddPCR’s superiority over qPCR in sensitivity and reliability. Normalized MYU levels demonstrated better specificity, negative predictive value (NPV), and positive predictive value (PPV) compared to serum PSA levels in predicting PCa. Additionally, a direct correlation between normalized MYU levels and PCa malignancy, indicated by the Gleason score, was observed.

#### 3.1.2. Bio-Rad QX ONE™ Droplet Digital PCR System (Bio-Rad Laboratories S.r.l, Milan, Italy)

A newer addition to Bio-Rad’s ddPCR portfolio, the QX ONE system is designed to offer enhanced sensitivity, throughput, and workflow efficiency compared to previous models [21]. This last article is a unique report in PubMed using this new technology: They report a new ddPCR-based assay to detect *Bartonella* spp. DNA in various sample types, including blood samples from patients with or without *Bartonella* spp. bacteremia. Results indicate that the ddPCR assay successfully detected 16 species/strains of Bartonella, including *B. henselae*, *B. quintana*, and others. Bartonella DNA was found in a previously negative sample, demonstrating 99% specificity. The ddPCR assay showed significantly better sensitivity, up to 0.5 bacterial genome copies per microliter of blood, compared to qPCR when testing patient blood and enrichment blood culture samples.

#### 3.1.3. Bio-Rad ThunderBolts™ Digital PCR System (Bio-Rad Laboratories S.r.l, Milan, Italy)

This platform offers a combination of high throughput and sensitivity for digital PCR applications. It features an automated workflow and advanced data analysis capabilities.

### 3.2. Digital PCR (dPCR)

Digital PCR is a broad term that includes various methods for partitioning a sample into discrete units (or partitions) to perform PCR amplification. In dPCR, the sample is partitioned into individual reaction vessels or compartments, each containing a single molecule or few molecules of the target nucleic acid. After PCR amplification, the presence or absence of the target molecule in each partition is determined using fluorescent probes or dyes that bind to the amplified DNA, allowing for absolute quantification of the target. This technique includes methods such as microfluidic-based dPCR [22], chip-based dPCR [23], and emulsion-based dPCR [24].

#### 3.2.1. Thermo Fisher Scientific’s QuantStudio 3D Digital PCR System (Thermo Fisher Scientific Inc., Waltham, MA, USA)

Thermo Fisher provides a digital PCR platform that enables absolute quantification of target nucleic acids. It offers high sensitivity and precision, making it suitable for applications such as rare mutation detection and gene expression analysis. Guo et al. [25] also proposed an interesting comparison between this technology and the BioRad ddPCR, demonstrating the high degree of consistency between the two techniques for the quantitative assessment of mutation in the plasma of circulating free DNA (cfDNA). Still, Dioni et al. [26] discussed the employment of this technology for the detection of R of the SARS-CoV-2 N protein (Nucleocapsid) and S protein (Spike), quantifying from 5.10 × 10^−1^ copies/μL to 79.76 copies/μL for N gene and from 1.37 copies/μL to 41.95 copies/μL for S gene.

#### 3.2.2. Stilla Technologies’ Naica™ System (Stilla Technologies, Villejuif, France)

Stilla’s Naica System utilizes digital PCR technology with innovative crystal-based microfluidics. It offers high throughput and sensitivity, allowing for precise quantification of nucleic acids across various applications [27]. Cornè et al. used this technology to demonstrate how due to its high sensitivity, robustness, and cost-effectiveness, dPCR is well-suited for clinically detecting ERBB2 mutations compared to next-generation sequencing (NGS) approaches [28]. In these terms, their work is very explanatory as they not only analyze 4 different mutations of the ERBB2 gene and the wild-type genotype but also correlate the measured concentration with the expected one, ranging from less than 100 copies/PCR to more than 1000, obtaining R^2^ values not lower than 0.9948.

#### 3.2.3. QIAcuity Digital PCR System by QIAGEN (QUIAGEN, Hilden, Germany)

A powerful tool for molecular diagnostics, research, and other applications requiring precise quantification of nucleic acids [29]. It can analyze multiple targets simultaneously in a single reaction, increasing efficiency and saving time. Tumpach et al. [19] showed that the QIAcuity dPCR platform allows for sensitive and accurate quantification of both intact and defective proviruses. The QIAcuity dPCR had low intra- and inter-plate variability, was linear over a dilution of 4 logs, and could detect HIV DNA down to 1–10 copies. The system is discussed thoroughly in the next paragraph.

#### 3.2.4. Digital LightCycler^®^ System Developed by Roche Diagnostics (Hoffmann-La Roche, Basilea, Switzerland)

With a combination of 3 nanoplate configurations, 6 advanced optical channels, and master mixes of DNA and RNA at a higher concentration than other platforms, the Digital LightCycler can support the laboratory from the publication of its research to the development of tests of clinical value.

#### 3.2.5. Lab on an Array (LOAA) Digital Real-Time PCR Analyzer System by OPTOLANE (OPTOLANE Technologies Inc., Giheung-gu, Republic of Korea)

LOAA is a PCR system that is divided into wells that use a semiconductor called a CMOS photo sensor, which allows for multiplex PCR, real-time digital PCR, and real-time PCR on the same platform. The workflow for LOAA Digital PCR is the simplest. One system can be used to conduct both the PCR and the analysis. No additional instruments are needed, such as a well partitioning instrument, droplet generator, or additional heat cycler. A finely divided well of the cartridge can accommodate each individual PCR reaction, and samples for analysis can be loaded into the well by simply injecting them. An interesting comparative paper [30] reported identification experiment using the Stilla Naica System, Droplet Digital PCR Technology, and LOAA Digital Real-Time PCR Analyzer System among commercially available dPCRs confirming that all equipment has the potential as a platform for species identification. In particular, compared to other dPCR equipment, the LOAA Digital Real-Time PCR Analyzer System has been made smaller and lighter with the manufacturer’s proprietary technology.

### 3.3. Technical Features

There are several dPCR or ddPCR platforms available, each offering unique features and capabilities for various applications in molecular biology, genomics, and diagnostics. A general comparison of the main three technologies is reported in Table 2. It is intriguing to note how the dPCR assay has been shown to improve the consistency of quantitative results across labs [31]. Nowadays, the most prominent digital PCR platforms include the following:

#### 3.3.1. Bio-Rad QX200™ Droplet Digital PCR (ddPCR) System

The QX200 Droplet Digital PCR System is made up of two instruments: the QX200 Droplet Generator and the QX200 Droplet Reader, as well as the consumables they require. The QX200 Droplet Generator is used to divide the ddPCR reaction mixture into thousands of nanoliter-sized droplets. Each droplet acts as an individual reaction chamber, allowing for absolute quantification of nucleic acids without the need for a standard curve. The system utilizes microfluidics technology to partition the sample and reagents into uniform droplets. This ensures precise and reproducible partitioning, leading to accurate quantification results. Following droplet generation, PCR amplification is carried out in each droplet individually. The system uses standard thermal cycling conditions for PCR amplification. After amplification, the droplets are read using a two-color detection system, allowing for the discrimination between positive and negative reactions. After PCR on a thermal cycler, each sample’s droplets are individually analyzed on the QX200 Droplet Reader. Droplets are easily read because they are streamed as a single file through a two-color optical detection system in serial. A maximum of 96 samples can be processed per run. The PCR-positive and PCR-negative droplets are counted to provide an absolute quantification of target DNA in digital format. Alternatively, amplified products can be extracted from droplets after PCR for further processing, such as sequencing or cloning. This system and its technology have been used for the detection of *Erwinia amylovora* viable Nonculturable Cells in Apple Bark [32], copy number variation in *FCGR3B* [33], and also for ctDNA detection in patients with early-stage breast cancer [34].

#### 3.3.2. QIAcuity Digital PCR System by QIAGEN

Similar to other digital PCR systems, the QIAcuity system utilizes microfluidics technology to partition the sample and reagents into individual microfluidic chambers containing a single or few copies of the target nucleic acid. Sample preparation, like qPCR experiments, begins with the transfer of master mix, probes, and primers to a 96-, 24-, or 8-well nanoplate, followed by the addition of the samples. Each well can contain 26,000 or 8500 partitions. The system combines partitioning, thermocycling, and imaging into a single fully automated instrument that allows users to go from sample to result in under 2 h. The Software Suite (v.2.5.0.1) allows for analysis, including the concentration of your target sequence in copies per microliter, as well as quality control measures such as positive samples or NTC. This analysis can also apply to remote computers on the same local area network (LAN). All the available systems are reported in Table 3. Although it is not indicated as a diagnostic device, QIAcuity has wide implications in the field of research: It has been used for the quantification of miRNAs as biomarkers in disorders of consciousness [35], for the detection of SARS-CoV-2 in air sample [36] and for the detection of Histone H3 K27M somatic variant in cerebrospinal fluid [37]. It is interesting to note the way that QIAcuity and BioRad technologies are often compared, and in some studies, the results from both are reproducible [37,38], while in others, QIAcuity technology seems to be more sensitive to some targets like cfDNA [39].

#### 3.3.3. Digital LightCycler^®^ System Developed by Roche Diagnostics

The Digital LightCycler^®^ System provides a semi-automated workflow for polymerase chain reaction (PCR)-based nucleic acid testing (NAT). The system conducts digital endpoint PCR analysis on microfluidic partitions, combining instrumentation, consumables, reagents, and data management to provide an efficient workflow from sample partitioning to result interpretation. The Digital LightCycler^®^ System is a semi-automated system for detecting and quantifying nucleic acid target copy numbers in vitro, designed for professional use in diagnostic and screening laboratories. The system consists of the Digital LightCycler^®^ Analyzer, the Digital LightCycler^®^ Partitioning Engine, and the required consumables, core reagents, and application software. All assays are designed independently of the Digital LightCycler^®^ System. The hexagonal-shaped nanowell partitions and the filling dye that validates partitioning enable the Digital LightCycler^®^ System to handle a wide range of extracted samples, including formalin-fixed paraffin-embedded tissue (FFPET), cfDNA, whole blood, and others (Table 4). Target sequences are effectively concentrated inside separate microreactors thanks to these partitions. This concentration effect reduces template competition, allowing for the detection of rare mutations and a greater tolerance to inhibitors found in a sample.

## 4. Digital PCR in Water Microbiology

Despite global improvements in hygiene levels and healthcare systems, part of the world’s population still does not have access to clean, safe, and drinkable water. According to the last report of the United Nations, this proportion is about 26% of the world’s population (2 billion individuals) mostly belonging to low-income countries [40].

Water represents a vehicle for the transmission of a plethora of pathogens responsible for the so-called waterborne diseases. Animal and human feces are among the main sources of contamination, and the fecal–oral route is the most common transmission pathway mainly due to direct ingestion of or to the use of contaminated water for washing foods [41,42]. Enteric viruses, including adenoviruses, enteroviruses such as coxsackieviruses and polioviruses, hepatitis A and E viruses, rotaviruses, astroviruses, and noroviruses, are the most prevalent in the feces of infected individuals and exhibit the highest persistence in aquatic environments, making them relevant for human health [43]. Similarly, concerning bacteria, water contamination mainly stems from fecal sources. They include not only *Escherichia coli* and *Enterococcus* spp. but also *Vibrio cholerae*, *Staphylococcus aureus* as well as the genera *Salmonella*, *Campylobacter*, *Shigella*, *Serratia*, *Enterobacter*, *Proteus*, *Klebsiella*, *Citrobacter*, and *Yersinia* [14,42]. Among aquatic environments implicated in pathogen transmission, wastewater stands out as a remarkable example, being a multifaceted matrix originating from domestic, hospital, agricultural, and industrial activities. Wastewater treatment plants (WWTPs) are designated to remove organic and inorganic contaminants from wastewater; nevertheless, sometimes the disinfection is not completely effective making them a reservoir for numerous infectious agents, a fertile ground for recombination events, and a route of transmission between different compartment of the society [44,45]. Monitoring both the presence and the persistence of pathogens in water sources is of paramount importance and requires the application of robust and reliable technologies able to provide unambiguous outcomes. Within this framework, digital PCR represents an outstanding technology whose major advantages lie in performing absolute quantification and bypassing PCR inhibitors commonly found in water and wastewater, ensuring accurate detection even in complex matrices [46,47]. During the pre-COVID era, qPCR represented the gold standard molecular technology in the water microbiology field, including wastewater surveillance [48,49,50,51,52,53,54]. However, due to its promising features, digital PCR has been increasingly employed within the last decade, and several studies have been published evaluating its effectiveness. Most of them revealed equal or higher precision [54], sensitivity [55,56], and reliability compared to qPCR, as well as its lower susceptibility to anthropological inhibitors [54,55,56]. Additionally, as evidenced by Varela et al., dPCR offers a resolution to the issue of overestimating targets in RT-qPCR associated with relative quantification and the need for a standard curve [57].

### 4.1. dPCR in SARS-CoV-2 Wastewater-Based Epidemiology

The COVID-19 pandemic led to a remarkable increase in dPCR technology employment, especially in wastewater-based surveillance, as highlighted by the exponential rise in the number of publications [14]. The majority of them tracked the regional presence of SARS-CoV-2 over time, primarily aiming to evaluate the effectiveness of RT-dPCR/ddPCR in detecting the virus and to compare wastewater-based epidemiology (WBE) data with clinical findings [58,59,60,61]. A strong correlation was found between the two approaches. Moreover, several studies showed that SARS-CoV-2 abundance in wastewater influents usually reached the peak slightly earlier than the clinical outbreaks making WBE a vital sentinel system for counteracting COVID-19 clinical outcomes [62,63,64,65,66]. Izquierdo-Lara et al., instead, employed reverse transcription droplet digital PCR (RT-ddPCR) to investigate Alpha, Delta, and Omicron BA.1 and BA.2 variants of concern occurrence in Rotterdam. Although the method shows good reliability, as broadly known it is a targeted approach therefore useless for the identification of new variants, for which whole-genome sequencing is strictly required [67]. Lastly, dPCR has also been applied to assess the effectiveness of WWTPs in SARS-CoV-2 removal [68].

### 4.2. dPCR in Wastewater Monitoring

The effectiveness proved by dPCR in the context of SARS-CoV-2 WBE highlighted its importance and promoted its use also for monitoring other pathogens.

Mtetwa et al., for example, used ddPCR to evaluate the presence of ARGs related to tuberculosis resistance in WWTPs in South Africa. ARGs associated with aminoglycosides and, interestingly, with the new drug bedaquiline have been detected [69]. The same research group also determined the concentration of both total *Mycobacteria* and members of the *Mycobacterium tuberculosis* complex, showing the low impact of treatment in their removal [70]. More recently, the abundance of *sul*2 (resistance to sulfonamides) and *tet*W (resistance to tetracycline) was assessed in urban water and wastewater distribution systems in Alicante (Spain). Similar results have been reported for both ARGs with higher abundances in hospital wastewater and WWTP influents [71,72].

### 4.3. dPCR in Surface Water Monitoring

In a similar way, in the field of surface water monitoring, the number of studies using dPCR has rapidly increased. Staley et al. assessed the efficacy of dPCR in measuring *E. coli* and microbial source tracking markers (MST) in the Rouge River watershed and Rouge Beach (Great Lakes area, USA). They discovered that human fecal contamination along Rouge Beach and in the lower portions of the watershed was mostly related to rain events, whereas gull fecal contamination was shown to be the main kind of contamination at the beach [73]. A study by Harringer et al. aimed to evaluate Comammox (COMplete AMMonia Oxidizers) concentration in 13 lakes across Switzerland, Austria, and Germany. Using a ddPCR assay to detect the *amo*A gene, they revealed the propensity of *Nitrospira* to live in sediment/water surface interface rather than in pelagic zones [74]. Pendergraph et al. employed ddPCR to detect fecal contamination in water sources in Absaroka Beartooth Wilderness (USA) which, despite being remote, became a popular destination for human visitation. Although this could potentially affect water quality, this study suggested a low prevalence of water-borne pathogens such as *E. coli* and human-derived *Bacteroides* [75]. Interestingly, a study by Vezzulli et al. rejected the hypothesis considering Tanganyika Lake (Africa) a reservoir of *V. cholerae*, as observed for other water sources in endemic cholerae regions. Specifically, by combining ddPCR and targeted genomics they identified *V. cholerae*; on the other hand, *V. cholerae* O1 and O139 toxigenic strains were not detected, possibly due to the low salinity of the lake [76]. Furthermore, ddPCR has recently been applied to ARGs’ monitoring in river and marine waters. A study by Coertze et al. identified the presence of AmpC β-lactamase and *intI*1 genes in genomic and plasmid DNA from South African rivers, highlighting a potential dissemination risk of antibiotic resistance through horizontal gene transfer [77]. Similarly, Di Cesare et al. revealed the long-term occurrence of *sul*2 (48%) and *intI*1 (76%) in plankton marine samples [78].

### 4.4. dPCR in Drinking Water Monitoring

Ensuring communities have access to safe drinking water is essential to promote ongoing and sustainable social and economic advancement [79]. Recent studies employed ddPCR to monitor the presence of pathogens, including bacteria, viruses, and protozoa, in order to assess drinking water quality. Spencer-Williams et al., for instance, performed species-specific ddPCR assays to investigate microbial growth and microbial ecology changes across American drinking water distribution systems (DWDS) due to the use of orthophosphate (PO_4_)^3−^. The latter is commonly used to counteract lead corrosion; however, this study confirmed its involvement in increasing the total number of bacteria and altering the microbial community. Specifically, an increase in Nontuberculous Mycobacteria (NTM) and a decrease in *Legionella pneumophila* were registered [80]. Logan-Jackson et al., instead, screened a complete drinking water system and cooling towers for the cooccurrence of different *Legionella* species and their natural protozoan hosts (*Acanthamoeba* spp. and *Naegleria fowleri*) using ddPCR. Overall, this study revealed a significant cooccurrence of *L. pneumophila*, *L. micdadei*, and *L. bozemanii* and *N. fowleri* [81]. A study by Kitajima et al. focused on the assessment of microbial abundance in both biofilm and water samples from a chloraminated DWDS. Specific ddPCR assays targeting bacterial and archaeal 16S rRNA genes, *Mycobacterium* spp., ammonia-oxidizing bacteria (AOB), and *cyanobacteria* were used. In contrast to previous studies, they did not show differences between planktonic and biofilm communities [82]. Lastly, Bivins et al. used ddPCR to evaluate water quality in continuous water supply (CWS) and intermittent water supply (IWS) systems in Nagpur, India. As expected, bacteria such as *Shigella*, enteroinvasive *Escherichia coli* (EIEC), enterotoxigenic *E. coli* (ETEC); protozoa like *Cryptosporidium* spp. and *Giardia* spp.; viruses like norovirus GI and GII and adenovirus A-F resulted less prevalent at household taps served by CWS compared to those served by IWS [83].

## 5. Clinical Diagnostic Application

Sepsis is a life-threatening condition triggered by the body’s dysregulated response to infection and poses a significant challenge in critical care medicine [84]. It often arises as a complication of severe trauma, burns, shock, or infection, affecting critically ill patients both medically and surgically. With the potential to progress to septic shock and multiple organ dysfunction, sepsis stands as a leading cause of global health loss, characterized by high mortality and morbidity rates [85]. dPCR stands out for its ability to precisely identify both pathogenic microorganisms and drug-resistant genes at an early stage [86,87,88,89,90], offering promising avenues for improving sepsis management. Studies evaluating dPCR’s efficacy in detecting pathogenic microorganisms associated with sepsis demonstrate its exceptional sensitivity and specificity in early pathogen detection and timely treatment initiation. Zheng et al. found significant differences in SOFA scores and 28-day mortality between patients identified solely by blood culture and those identified solely by digital PCR, with the latter group exhibiting milder symptoms typical of bloodstream infection [91]. Chen et al. observed rapid recovery in neonatal samples, initially negative via blood culture and qPCR, following antibacterial treatment guided by dPCR diagnosis, highlighting dPCR superiority in accuracy and potential in clinical settings by identifying a higher number of positive samples compared to other traditional methods [92].

Its non-invasive nature and capacity to detect drug-resistant genes make it a valuable tool in guiding treatment decisions and mitigating antibiotic misuse. With heightened sensitivity, precision, and tolerance to inhibitors, dPCR offers advantages over traditional PCR methods, particularly in scenarios where blood culture sensitivity may be compromised.

Recent studies have demonstrated dPCR’s application for diagnosing lower respiratory infections, such as ventilator-associated pneumonia (VAP). This application holds significant potential but comes with unique challenges [93]. In the lower respiratory tract, dPCR’s high sensitivity and specificity make it useful for detecting bacterial DNA in bronchoalveolar lavage (BAL) samples [94]. This capability is particularly beneficial for diagnosing VAP, where distinguishing between colonization and infection is crucial. The technology also facilitates the rapid detection of antibiotic resistance genes directly from respiratory samples, enabling timely and appropriate antibiotic therapy, which is essential for critically ill patients. However, the complex microbial environment of the lower respiratory tract poses challenges to dPCR application [95]. The presence of commensal organisms can complicate result interpretation and increase the risk of false positives [1]. While dPCR is robust against inhibitors found in non-sterile samples, this issue is not entirely eliminated. In comparison, dPCR application in sterile sites, such as blood, is more linear due to the simpler microbial background, which reduces contamination risks and enhances diagnostic accuracy [96]. Further research and clinical validation are needed to optimize dPCR protocols for non-sterile samples from the lower respiratory tract. Establishing standardized procedures through clinical studies will improve the reliability of dPCR in these complex environments. Implementing dPCR in routine clinical practice involves significant costs for specialized equipment and training, but its benefits in diagnostic accuracy and speed, especially in critical care settings like VAP management, may justify these investments [93,97]

The challenge posed by drug-resistant bacteria in clinical diagnosis and treatment is significant. Digital PCR has the capability to detect viral resistance mutations with a detection limit of 0.1% abundance, surpassing qPCR, which is commonly employed in clinical settings, by over 50-fold [98]. Furthermore, dPCR technology offers advantages over traditional PCR, as it does not rely on a standard curve, boasts higher sensitivity and accuracy, and exhibits greater tolerance to inhibitors, resulting in enhanced precision and reproducibility in detecting pathogenic microorganisms [46]. Moreover, dPCR outperforms blood culture in detecting potential pathogens in suspected patients with bloodstream infection (BSI), particularly benefiting those who received empirical antibiotic treatment [99]. Elevated levels of white blood cells, procalcitonin, and C-reactive protein correlate with higher pathogen copy numbers detected by dPCR [99]. The capacity to monitor pathogen loads and disease progression bears clinical significance, potentially enhancing antimicrobial regimens and guiding precision treatment therapy. Early administration of appropriate antimicrobial drugs in sepsis significantly reduces patient mortality [100,101,102], with rapid diagnosis and treatment potentially saving up to 80% of patients with septic shock [103]. dPCR is expected to yield accurate results for pathogenic microorganism infections an average of 2–3 days earlier than conventional methods, demonstrating clinical utility.

Despite its potential, the widespread adoption of dPCR encounters obstacles, primarily attributed to its cost. Sepsis is identified as the most expensive medical condition to address, contributing to the highest hospitalization costs not only in the United States [104] but also in European countries [105,106,107], thus presenting a substantial financial challenge. Implementing rapid digital PCR detection may raise testing expenses; however, the benefits it offers, including expected reductions in anti-infective treatment duration, hospital stays, and drug costs, justify its use. It would enable patients to receive prompt and precise treatment, improving prognosis, reducing complications, and minimizing patient readmissions, ultimately resulting in reduced overall costs. Several studies confirm the cost-effectiveness of reducing mortality and healthcare costs through rapid and accurate detection of pathogenic microorganisms [108,109]. Predictive models justified PCR’s added costs by its potential to improve patient outcomes and reduce healthcare expenses, particularly in settings with high treatment costs and significant risks of complications due to inadequate treatment [110]. Underscoring its potential to improve patient outcomes and reduce healthcare expenses, dPCR emerges as a valuable tool in diagnosing and managing sepsis effectively. Despite these challenges, continued research and refinement of dPCR hold the promise of revolutionizing sepsis management and enhancing critical care practices.

## 6. Future Prospects of dPCR

After having examined the extreme sensitivity, precision, and quantitative capabilities of dPCR in various fields, from microbiological diagnostics to environmental surveillance, we focused on understanding its potential and future perspectives.

This field is rapidly advancing with efforts focused on enhancing technique performance and reliability. Research applies in the development of highly specific and sensitive fluorophores to improve signal discrimination from background noise [111]. Future trends in dPCR focus on methodological optimization and on the integration of innovative technologies to enhance performance using, for example, advanced fluorophores to achieve higher resolution and sensitivity in analyses. Schlenker et al. proposed a method for virtual fluorescent color channel creation in dPCR devices using universal reporter molecules, expanding the standard color channels and offering excellent sensitivity for mutation detection. This approach is cost-effective and versatile, holding promise for applications like cancer diagnostics. Future studies may explore integrating readout devices and extending the concept to other techniques [112].

Additionally, the introduction of advanced algorithms for artifact correction, such as those based on machine learning, is revolutionizing dPCR data analysis, enabling more accurate target copy quantification. Yan et al. introduced an image-to-answer algorithm tailored for dPCR image processing, enhancing partition localization, F value extraction, and subsequent analysis. The algorithm, exploiting techniques like the “Hough transform and 3D projection transformation”, achieves precise partition identification and fluorescence intensity correction. The algorithm ensures reliable performance in analyzing chip-based dPCR systems, offering a versatile solution for current and future platforms [113].

Moreover, Lee et al. presented a deep learning-assisted approach for absolute quantification of target DNA in dPCR analysis, effectively detecting and distinguishing positive droplets. The method’s scalability promises enhanced accuracy by increasing the number of droplets, positioning the machine learning algorithm as a valuable tool for precise digital analysis of various infectious diseases [114].

Furthermore, the application of dPCR to metagenomics is a rapidly growing area of research, with its ability to accurately quantify specific DNA sequences in complex samples proving invaluable for studying microbial diversity and identifying organisms, particularly in clinical settings [1].

In conclusion, dPCR continues to evolve with promising prospects, including the implementation of new technologies and expansion into new sectors like metagenomics. Innovative approaches aimed at improving sensitivity and accuracy pave the way for a wide array of diagnostic and research applications in biomedical and environmental fields.

## 7. Conclusions

Digital PCR (dPCR) stands as a transformative technology with immense potential to revolutionize both wastewater surveillance and clinical microbiology. Throughout this review, we have highlighted the key features and applications of dPCR in these critical fields. In wastewater surveillance, dPCR emerges as a powerful tool for the sensitive and accurate detection of microbial pathogens and indicators. Its ability to provide absolute quantification, even in complex environmental samples, offers unparalleled insights into the dynamics of microbial communities. Overall, even though cultural methods are cost-saving and easy to perform, they come with intrinsic limitations related to non-cultivable microorganisms as well as the time required for their growth [115,116]; this allowed molecular techniques to become increasingly used in clinical and environmental microbiology applications. Moreover, the scalability and cost-effectiveness of dPCR platforms make it feasible for routine monitoring, enabling timely interventions to safeguard public health.

In clinical microbiology, dPCR demonstrates remarkable utility in the detection and quantification of pathogens with high precision and sensitivity. Its ability to detect low-abundance targets in clinical specimens like blood samples holds promise for early disease diagnosis, detecting antibiotic resistance genes and monitoring treatment efficacy. Furthermore, the ability of dPCR to multiplex assays enhances its efficiency in diagnosing polymicrobial infections and tracking multiple pathogens simultaneously.

Despite its numerous advantages, challenges remain in the standardization of protocols, optimization of workflows, and the need for robust quality control measures. Additionally, cost considerations and accessibility issues may limit its implementation, especially in resource-limited settings. Looking ahead, continued advancements in dPCR technology, including improvements in assay design, instrument automation, and data analysis algorithms, hold the potential to address these challenges and further enhance the utility of dPCR in various fields. Collaboration between researchers, clinicians, and industry stakeholders will be crucial in realizing the full potential of dPCR and exploiting its capabilities to mitigate public health risks associated with microbial contamination in both environmental and clinical settings.

In summary, digital PCR emerges as a game-changing technology with the ability to revolutionize microbial detection and quantification in diverse applications. Its sensitivity, accuracy, and scalability make it a valuable tool for understanding microbial dynamics, diagnosing infectious diseases, and safeguarding public health. As we continue to overcome technical and logistical challenges, the future of dPCR in these fields appears promising, paving the way for more effective strategies in disease prevention and management.

## Figures and Tables

**Figure 1 diagnostics-14-01598-f001:**
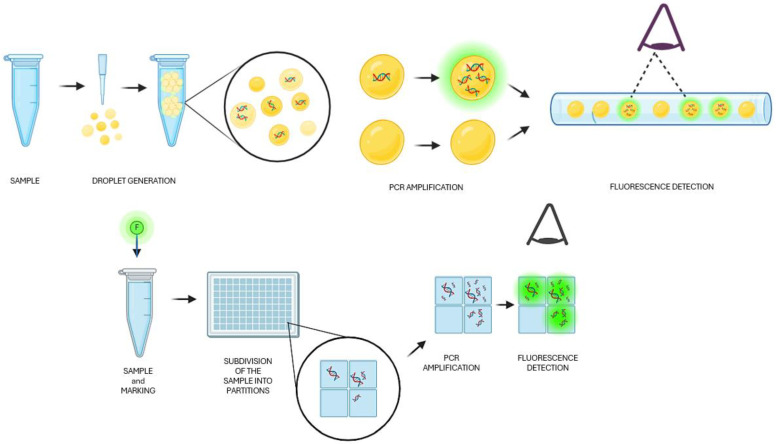
Graphical workflow of ddPCR (above) and dPCR (below). Independently created by an author (P.G.B.) using BioRender.com accessed on 25 May 2024, freely inspired by Kokkoris et al. [16].

**Table 1 diagnostics-14-01598-t001:** Comparison of dPCR and ddPCR.

Digital PCR (dPCR)		Digital Droplet PCR (ddPCR)	
PROS	CONS	PROS	CONS
dPCR encompasses various partitioning methods, including microfluidic-based, chip-based, and emulsion-based techniques, providing flexibility in experimental design.	Unlike ddPCR, dPCR techniques may have less precise control over droplet size and uniformity, potentially affecting assay performance.	ddPCR offers exceptional precision and accuracy in quantifying nucleic acids, particularly for low-abundance targets, due to the ability to analyze individual droplets.	ddPCR may have higher consumable costs compared to some dPCR platforms, especially for emulsion-based systems, which require specialized reagents.
Some dPCR platforms may have lower consumable costs compared to ddPCR, especially for microfluidic-based systems.	Certain dPCR platforms may have higher upfront costs compared to ddPCR systems, making them less accessible for some laboratories.	ddPCR can detect and quantify rare target molecules with high sensitivity, making it ideal for applications requiring the detection of low-level genetic mutations or rare transcripts.	Some ddPCR platforms have lower throughput compared to certain dPCR systems, potentially limiting scalability for high-throughput applications.
Certain dPCR platforms offer higher throughput options, enabling the analysis of a larger number of samples simultaneously.	While there are multiple dPCR technologies, the availability of commercial platforms may be more limited compared to ddPCR systems, leading to fewer options for researchers.	ddPCR typically requires smaller sample volumes and reagent amounts compared to traditional PCR or dPCR, potentially reducing experimental costs.	Manual handling of samples and droplet generation can introduce the risk of cross-contamination between samples, requiring careful experimental design and handling procedures.
In some dPCR systems, such as microfluidic-based platforms, sample handling is automated, minimizing the risk of contamination between samples.	Depending on the platform, dPCR may have lower sensitivity compared to ddPCR, particularly in applications requiring the detection of rare target molecules.	The compartmentalization of reactions in individual droplets can help mitigate the effects of PCR inhibitors present in complex samples, enhancing assay robustness.	ddPCR platforms may have limited options for integration with other analytical techniques, potentially restricting workflow customization for specific experimental needs.
dPCR platforms may be more amenable to integration with other analytical techniques, such as microfluidic-based sample preparation or downstream analysis.			

**Table 2 diagnostics-14-01598-t002:** Comparison of the main three available technologies.

Technology	Bio-Rad QX200™	QIAcuity	LightCycler^®^
Partitions	20,000	8500–26,000	20,000–28,000–100,000
Samples	Up to 96 for runtime	Up to 96 for runtime	Up to 96 for runtime
Concentration on MMx	2×, 4×	4×	5×
Channels	4, 6	2, 5	6
Working volume	20 µL	12 µL, 40 µL	15 µL, 30 µL, 45 µL
Medical devices	Yes	No	Yes
Price range	€€	€	€€€

Price range: € (low), €€ (moderate), €€€ (high)

**Table 3 diagnostics-14-01598-t003:** Technical characteristics of the three systems produced by QIAGEN for QIAcuity.

Feature	QIAcuity One	QIAcuity Four	QIAcuity Eight
Plates processed	1	4	8
Detection channels (multiplexing)	2 or 5	5	5
Thermocyclers	1	1	2
Run time	2 h	First plate approximately 2 h, every 80 min a following plate	First plate approximately 2 h, every 40 min a following plate
Throughput	Up to 384 (96-well) Up to 96 (24-well)	Up to 672 (96-well) Up to 168 (24-well)	Up to 1248 (96-well) Up to 312 (24-well)

**Table 4 diagnostics-14-01598-t004:** Technical features of the three plates available for the Roche LightCycler instrument.

Feature	High-Resolution Plate	Universal Plate	High-Sensitivity Plate
Working volume	15 µL	30 µL	45 µL
Partitions	100,000	28,000	20,000
Sensitivity grade	Copy number variation	Gene expression	Cell-free DNA
Some applications	NIPT, human genetic disease	Transplant rejection	Oncology, rare mutation detection

## Data Availability

Not applicable.

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
