# Peer review of "Advancing Pathogen Identification: The Role of Digital PCR in Enhancing Diagnostic Power in Different Settings"

_diagnostics, 2024, doi:10.3390/diagnostics14151598_

Round 1

Reviewer 1 Report

Comments and Suggestions for Authors

The manuscript submitted by Alessia Mirabile and all, entitled Advancing Pathogen Identification: The Role of Digital PCR in Enhancing Diagnostic Power in Different Settings, presents a comprehensive overview of the several commercial dPCR platforms developed in the last years, their characteristics, and their applications in infectious disease diagnostics and environmental surveillance.

The manuscript makes a good impression overall. The authors compare the main dPCR technologies available, highlighting each one's characteristics as well as its advantages and disadvantages.

I believe that the paper should be considered for publication in Diagnostics. The overview offered of this emerging and promising technology will attract those with closely aligned research interests.

Author Response

Reviewer 1

The manuscript submitted by Alessia Mirabile and all, entitled Advancing Pathogen Identification: The Role of Digital PCR in Enhancing Diagnostic Power in Different Settings, presents a comprehensive overview of the several commercial dPCR platforms developed in the last years, their characteristics, and their applications in infectious disease diagnostics and environmental surveillance.

The manuscript makes a good impression overall. The authors compare the main dPCR technologies available, highlighting each one's characteristics as well as its advantages and disadvantages.

I believe that the paper should be considered for publication in Diagnostics. The overview offered of this emerging and promising technology will attract those with closely aligned research interests.

We appreciate your positive feedback and are pleased that our comprehensive review has made a good impression.

Reviewer 2 Report

Comments and Suggestions for Authors

Overall, the study effectively captures the reader's interest, provides necessary information, and sets clear expectations for a comprehensive overview of the evolution of PCR technologies, culminating in an explanation of Digital PCR (dPCR). It successfully communicates complex concepts in molecular biology while maintaining a structured approach. To enhance further, simplifying complex sentences, improving transitions between paragraphs, offering specific examples of limitations, adding references, and ensuring consistent use of acronyms would be beneficial refinements.

Comments on the Quality of English Language

Some of the sentences are complex need to simplify the sentences. 

Author Response

Reviewer 2

Overall, the study effectively captures the reader's interest, provides necessary information, and sets clear expectations for a comprehensive overview of the evolution of PCR technologies, culminating in an explanation of Digital PCR (dPCR). It successfully communicates complex concepts in molecular biology while maintaining a structured approach. To enhance further, simplifying complex sentences, improving transitions between paragraphs, offering specific examples of limitations, adding references, and ensuring consistent use of acronyms would be beneficial refinements.

Comments on the Quality of English Language

Some of the sentences are complex need to simplify the sentences.

Thank you for your valuable feedback. We have simplified complex sentences and improved transitions between paragraphs. Additionally, we included examples of the limitations and revised the use of acronyms throughout the text.

Reviewer 3 Report

Comments and Suggestions for Authors

Authors produce a review on the application of digital PCR in infectious disease diagnostics and enviromental surveillance. The review is accurate and several concepts have been developed. If possible, I only would suggest to simplify a bit the english form. 

Comments on the Quality of English Language

A simplified english form could be helpful for the reader.

Author Response

Reviewer 3

Authors produce a review on the application of digital PCR in infectious disease diagnostics and enviromental surveillance. The review is accurate and several concepts have been developed. If possible, I only would suggest to simplify a bit the english form. 

Comments on the Quality of English Language

A simplified english form could be helpful for the reader

Thank you for your positive feedback on our review. We have revised the manuscript to simplify the language and ensure it is more accessible to a broader audience.